# What Type of Households in Mongolia Are Most Hit by COVID-19?

**Ganchimeg Gombodorj [1,\*]** and **Károly Pető [2]**

1. Karoly Ihrig Doctoral School of Management and Business, University of Debrecen, 4002 Debrecen, Hungary
2. Institute of Rural Development, Regional Economy and Tourism Management, Faculty of Economics and Business, University of Debrecen, 4002 Debrecen, Hungary; peto.karoly@econ.unideb.hu
* Correspondence: gombodorj.ganchimeg@econ.unideb.hu; Tel.: +36-20-5460-637

**Abstract:** The study aims to validate the measures taken by the government of Mongolia against COVID-19 and to analyse the negative impacts of COVID-19 on the livelihood of different types of households. The survey covers 362 households consisting of five different types, namely, apartment households, *ger* district households, herder households, vegetable growing households, and small farmer households. Principal component analysis is used to reduce the number of variables to a few factors that best explain the variation in the variables. Two to three components were found from the principal component analyses that describe (i) government policy, (ii) challenges due to COVID-19, and (iii) risk and vulnerability that occurred due to COVID-19. Multiple regression models attributed by the household's type were used to analyse the impact of the selected variables on the households' income. According to the regression results, herding households are the least affected, compared to the other four types of households. The government measures to mitigate the negative impacts of COVID-19 have better results for apartment households and *ger* district households. Rural households are less affected and seem to be more resilient to COVID-19 shocks than other households.

**Keywords:** COVID-19 shocks; government measures; household livelihood; rural households

## 1. Introduction

COVID-19 is a health shock that originated in China in December 2019. It is a virus that transmits disease due to the Severe Acute Respiratory Syndrome Coronavirus 2 (SARS-CoV-2), which has caused a wide array of damages to health, economy, and social life [1]. The impacts of COVID-19 have been enormous, and it is not easy to imagine the future impacts it will have.

Facing the spread of COVID-19, many countries have restricted their population movement, asking them to stay home safely [2], and governments took different strategies against it [3]. While these measures may help limit the spread of the virus, there are fears around the world that it could have unintended social and economic consequences for people, including the poor and most vulnerable people [4].

The COVID-19 shock and the containment measures would seriously impact poverty, health, and income [2,5]. Moreover, the COVID-19 pandemic has an adverse effect on the poor population, which are already at risk of being left behind [5].

The pandemic is still severe and even worsening in some countries. Mongolia was one of the first countries to take preventive measures, closing its borders, shutting down schools and businesses, restricting social gatherings, and banning international arrivals, thereby allowing the country to self-isolate and avoid a more significant outbreak [6]. On 10 March 2020, Mongolia was shocked by the news of its first imported case of COVID-19. Prior to the announcement of the first case, the government and the public were readied to make a scenario [7]. The government of Mongolia has adopted a phased self-quarantine

policy to prevent and control the pandemic, which has impacted every sector and the livelihoods of Mongolian households. Although the full effect of COVID-19 has not been identified, Mongolia felt numerous socio-economic effects: in the first season of 2020, the economic growth declined by 10.7% (year-on-year), driven by a 30% decline in the mining and quarrying sector, which accounts for 24% of the GDP. The service sector, including trade, which has the largest share in the GDP (39%), declined by 6.8% due to COVID-19 containment measures. However, the growth contributions were from the agriculture (14% growth) and non-mining manufacturing sector (6.4% growth) [8].

During the COVID-19 lockdown, providing food to urban and suburban citizens and ensuring herders' and producers' incomes became essential tasks for the government of Mongolia [9], where 28% of households (HHs) are below and 15% are just above the poverty line [8]. Of 908.7 thousand total households in Mongolia, 60 percent live in an apartment connected to the independent engineering system. The remaining 362 thousand households live in *gers* or tiny private self-built houses without a proper sanitation system [9]. A total of 72.2% of households (including 188 thousand nomadic herder HHs) live in a *ger*. Herder HHs live in *gers* regardless of their income levels because they are easy to construct. The remaining households live in privately built houses in soum centres. Vegetable grower households live near the capital and in crop-production areas of Mongolia, and are engaged in small-scale potato and vegetable growing businesses. Vegetable production shares 10 percent of the total crop sector, and 17,459 vegetable grower households are engaged in this sector [10]. In this study, the farming households refer to farmers who live on the city's outskirts and herd a small number of cows, selling milk and other dairy products.

The population density of the capital city is 339.8 people per square km. The western, Khangai, central, and eastern regions are sparsely populated, with 0.8–1.6 people per square km [10]. Urbanisation, population density, and many households not connected to the integrated engineering system contribute to high levels of soil and water pollution and infectious diseases in urban areas [9].

Poverty levels in urban and rural areas vary. According to the poverty survey in 2018, 25.9% of total households in Ulaanbaatar, 30.1% in *aimag* centres (the first level of the administrative unit of Mongolia), 28.9% in *soum* centres (the second level of the administrative unit of Mongolia) and 32.9% in rural areas are in poverty [11]. Herder households with few animals (poverty incidence rate 33.2%) are the most likely to be left behind. Kalinowski and Łuczak and Kerbage et al. [12,13] noted that the rural areas lack good health services and have poor sanitation infrastructure, poor social protection, and poor governance capabilities, and that most government activities are aimed at the urban areas, making the rural area more vulnerable and risk-prone to COVID-19.

However, Waibel et al. and Raju et al. [3,14] found that while rural households are at risk for COVID-19, they are more resistant to such shocks. Moreover, Malatzky et al. [15] found that city residents in Canada, USA, Australia, and Norway escape to the countryside because rural areas are relatively safe. Jia et al. [16] emphasise the advantages of rural areas, consisting of lower population density, more green nature, close social networks, and more self-subsistence capability, as it is possible to grow one's own food. Similarly, rural households are less likely to suffer from food shortages in rural Mongolia, where people are sparsely populated with an average population of one person/km2. They are self-subsistent and procure meat, milk, and vegetable products. For instance, 80% of the meat consumption of herder households come from their livestock [17]. In order to reduce the negative effects of COVID-19 and prevent other adverse effects, the government needs to develop and implement different policies that consider the specifics of urban and rural households. Luo et al. [2] emphasized that conducting timely research on whether the pandemic affects rural and urban households, on how the spread of the pandemic affects rural households, and on how they respond to it is necessary.

Hence, the research intended to analyse how COVID-19 affects different households, depending on where they are located. To achieve the research goal, the study attempted to answer the following research questions:

(1)　How are the measures taken by the government of Mongolia against COVID-19 affecting household livelihoods?
(2)　What is the impact of COVID-19 on the rural households' livelihoods, and do they cope with this shock?
(3)　To what extent are rural households suffering from this shock, compared to others?

The study hypothesised that rural households, such as Mongolian herders and vegetable growers, are less affected by COVID-19 than households in the capital city and other urban areas.

The data used in the article were collected from June–July 2021 to capture the livelihoods of Mongolian households when the pandemic was widespread and at its peak. At the time of the survey launch, in June and July 2021, there had been more than 1100 daily confirmed COVID-19 cases and 9–12 daily deaths registered in Mongolia [10].

The article structure is as follows. The second section presents a systematic literature review analysis on livelihood and COVID-19. The conceptual framework that highlights connections between shocks and households' livelihood is explained in the third section. The data and methodology are introduced after the conceptual part. The fifth section discusses the research results and includes the policy response measures taken by the government of Mongolia to mitigate the impact of COVID-19. Finally, the sixth section concludes the research results, including recommendations and the study's limitations.

## 2. Systematic Literature Review

Although research has been carried out on the various household shocks, the COVID-19 pandemic is a new public health shock. Even though COVID-19 is new, we conducted a systematic literature review in October 2021 using bibliographic records in the Scopus database; under filtering, the keyword "COVID-19" screened 283,344 documents published between 2019–2021. After refining the subject area to "social science and economics and econometrics and finance", the type of the document to "research article", and the language to "English only", 2295 documents remained. The keyword "livelihood" was added to further narrow the search to be in line with our research objective, resulting in 135 research articles. "COVID-19", "livelihoods", "governance", and "lockdown" were the most repeated keywords (Figure 1). The United States of America led the number of publications related to COVID-19, followed by India, the United Kingdom, and Australia.

In addition to 135 research articles, 53 documents issued by the World Bank, the Food and Agriculture Organization, and the government of Mongolia were added and analysed by the free systematic literature review software tool, Rayyan. After detecting duplicate steps, 17 documents were removed. After analysing the search results, 49.1% of the documents not relevant to our study were excluded, 10.3% of the documents were classified as maybe, and 40.6% of the documents (76) were selected for further detailed review for a conceptual framework.

Research related to COVID-19 in Mongolia are few and from different aspects. Ganbayar [18] analysed Mongolian external debt and finance issues during the early spread of COVID-19. Other studies assessed the Government of Mongolia's early responses and measures against COVID-19 [6,9,19,20]. Some studies analysed the impact of the pandemic on education and online distance education [21,22]. The remaining studies were related to medical science: work stress among nurses and medical doctors and clinical COVID-19 testing [23–26]. One study was focused on the pandemic's impact on livelihood; however, it was limited to only Darkhan, the third-largest city in Mongolia. All the above studies were conducted in early 2020, when Mongolia did not yet have an internal COVID-19 case, only an imported one from abroad. The first internal COVID-19 case was registered on 10 November 2020.

After reading the articles and documents found as a result of the systematic literature review, the conceptual framework was developed.

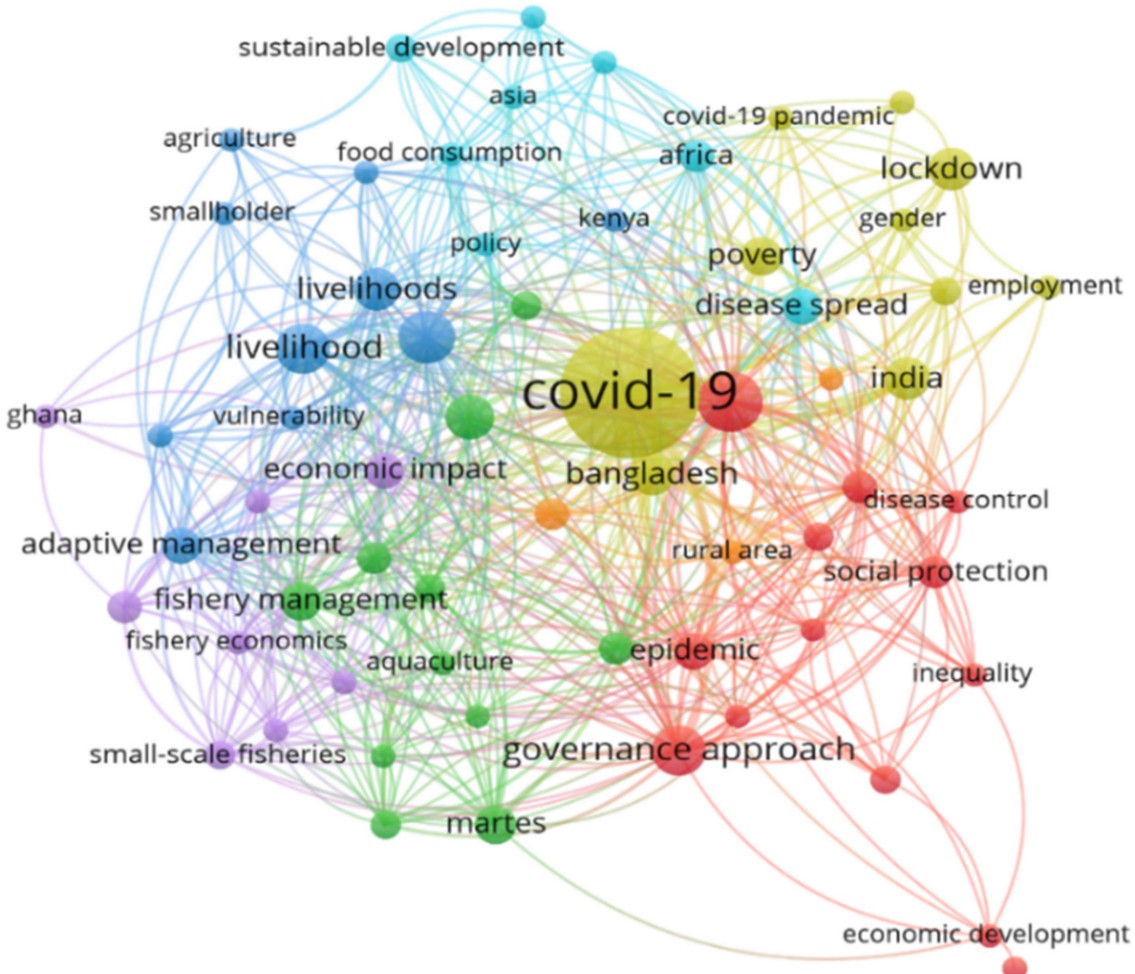

**Figure 1.** Selected articles' keywords and their connections. Source: Authors' construction, using VOS viewer (free software).

## 3. Conceptual Framework

Poverty depends on vulnerability context factors such as shocks, trends, and seasonality [27]. Ansah et al. [28] defined a shock as any event that may disrupt the normal functions of socio-economic agents and/or their activities, impose challenges, and threaten household food security. Shocks such as health, natural disasters, economic, and livestock health, and civil conflict can directly or indirectly influence assets and livelihoods.

The COVID-19 pandemic is a significant health event that has an impact on households' livelihood outcomes. In addition to the direct health effects of COVID-19, the administrative decisions taken against it affect production and households in many ways; for example, access to education [29] and health care, deteriorating education [30], limited market access due to quarantine and lockdown [29,31–34], and the reduced availability and high price of goods [29] are all related to the effects of COVID-19 and the steps taken to combat it. The risks to agriculture are increased production costs due to the lack of agricultural workers in the production stage [30], a loss of efficiency due to blocked transportation in the logistics stage, and a decline in sales volume and prices in the market stage [30,31,35–37].

Coping strategies are important and depend on the shock characteristics [28]. Households aim to preserve productive assets and diversify their activities for the short term. Ashraf and Routray [38] reported that decreasing consumption, skipping meals, and relying on social networks are possible coping strategies against shock.

Countries have adopted their own government policies and strategies to overcome COVID-19. A country's anti-COVID-19 measures, both domestically and globally, could

have a direct impact on incomes and threaten its citizens' livelihoods, many of whom are already members of economically vulnerable households. In response, governments continue to take various measures to ensure the economic security of households [39]. Australia, Canada, United States, India, Germany, Italy, UK, China, Indonesia, Japan, and Malaysia took measures on electricity bills, such as halting any administrative changes, suspending and discounting charges, and sometimes even offering the service for free. Argentina and France suspended all utility bills for certain periods [40]. Brazil, Mexico, South Africa, and South Korea provided financial, social, and economic aid to their citizens [40]; cash transfer programs were implemented in Thailand, Vietnam [41], and the United States [42]. Bui [41] found that financial support to households not only increases spending but also encourages positive trust in the government and improves mental well-being, based on an efficiency study of the cash transfer program for eligible individuals and poor in Vietnam and for farmers in Thailand for three months, with support ranging from USD 34 to USD 241. As another example, in the United States, consumption and spending increased immediately when households received cash assistance from the government, and households tended to spend more on durable goods [42]. According to Georgarakos and Kenny [43], in European areas, governments provided extensive financial support to businesses and individual households. These measures include pandemic-related financial support and subsidies, including in-kind support (e.g., via extended childcare), employment subsidies loans, payment guarantees, and moratoria. The study assessed that even though 70 percent of households had not received government support, people were favourable to higher spending [43]. In addition, loan deferring or decreasing interest rates supports the household economy. Di Maggio [44] discovered that monthly mortgage loan repayments declined by up to 50%, motivating the consumption of durable goods, which increased by up to 35%. Malaysia distributes e-money to its citizens, delays loan repayments, reduces some types of taxes, offers food and shelter for seniors and wage subsidies, and provides support to frontline workers and rural households. Shah et al. [45] assessed that these measures were rated as the most effective measures taken by the government of Malaysia to initiate people-based economic growth.

Households facing health shocks, such as family members becoming ill, may have to sell their livestock or use their savings to cover the resulting costs [46–48]. Ansah et al. [28] stated that households respond differently to shocks depending on their level of assets. In other words, poor households are hit harder by the negative impact of any type of shock. Moreover, coping strategy decisions depend on the shock characteristics related to its essence [49]. Illness and death of a household member reduce household labour allocation and increase health or funeral expenditures [50]. Drought, floods, crop pest infestation, animal diseases, and high food/input prices might cause a different shock. The shock could affect a single household (idiosyncratic) or many households in a given location (covariate).

Moreover, the severity of a shock and its duration are essential characteristics to define. Whether in isolation or coincidentally, shocks lead to actual income loss by reducing profits, increasing consumption costs, or destroying assets [51]. If a shock is individual or mild, the coping strategies a household chooses so as to minimise the effects on its welfare depend largely on the associated costs. If shocks are intense, they cause higher costs, and households would be forced to adopt different coping strategies. Moreover, multiple shocks and their combined reinforcing effects on welfare cause vulnerabilities [28].

Ellis [52] points out that asset depletion often seems to be the last option when households experience shocks with high impacts. Moreover, he outlines five main coping mechanisms that are adopted sequentially when households face shocks that threaten food security. The first is anticipatory, involving income diversification; the second draws on social networks. If these two mechanisms are insufficient for coping, the next is for some household members to migrate temporarily. In addition to migration, households may deplete agricultural assets, such as livestock. If all these mechanisms fail, the last option is to deplete fixed assets, such as land or buildings.

Wealthier households with more assets tend to be more resilient and cope better with shocks. However, the specific assets used for coping depend on the severity of the shock [28].

When shocks coincide, they may interact with and reinforce each other, producing a combined effect that differs from the sum of the isolated shocks [28,49]. Ansah et al. [28] called this additional effect an incremental effect. However, if savings are insufficient, the household may be forced to use additional strategies until all available options are exhausted and before productive assets need to be depleted [28]. Assume that COVID-19 case hit one household while the other is healthy: the food price and everyday consumption are raised by nationwide lockdowns, increased transaction costs, and poor logistic systems. In this case, market price shock interacts with each household, and the household hit by COVID-19 has caused a cumulative effect.

## 4. Materials and Methods

### 4.1. Study Area

Mongolia is a lower–middle-income country with 3.4 million inhabitants that neighbours Russia and China. Mongolia has 1.5 million square kilometres, with frequent extreme weather events, hot and dry summers, and freezing winters. A majority (70%) of the area is rangeland [10], which supports the traditional nomadic livelihoods and other sectors [53]. According to the regional development concepts approved by the parliament of Mongolia in 2001, the country has been divided into five different economic regions: western, Khangai, central, eastern, and Ulaanbaatar. By the end of 2020, 69% of the total population lived in urban areas, while 31% lived in rural areas, moving around the country with their livestock year round. A total of 47.6% of the total population lives in the capital city Ulaanbaatar.

### 4.2. Data Collection

The survey covered 362 households from five different economic regions. Households of 16 *soums* of eight *aimags* and six capital city districts are selected (Figure 2).

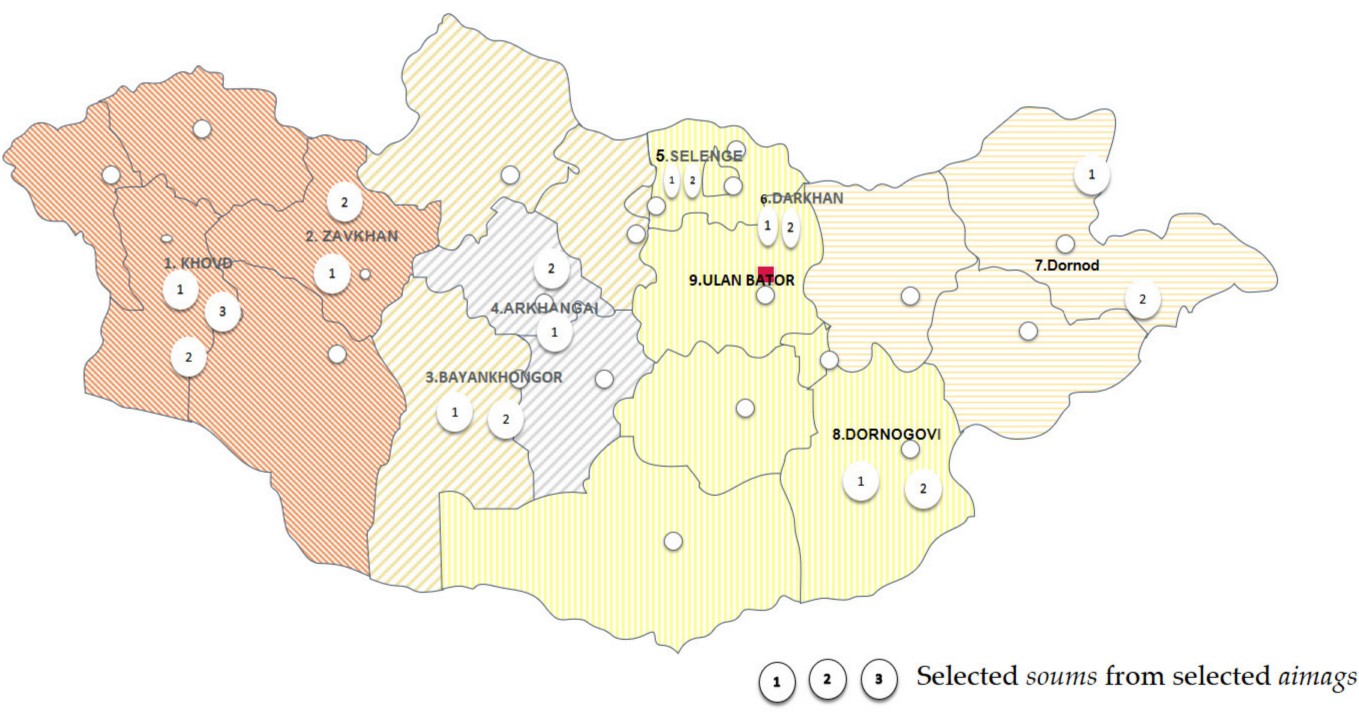

**Figure 2.** Selected study areas from the economic regions in Mongolia.

To compare the impact of COVID-19 on the different types of households' livelihoods, we classified households into (i) living in *ger* districts (Mongolia's traditional housing; most poor households live in the capital city), (ii) apartment living households (with central heating and sanitary systems), (iii) vegetable grower households (majority are living in the northern part of Mongolia), (iv) herder households (rural households with livestock), and (v) small farmers (living near the capital city with small dairy businesses).

In Mongolia, the first internal COVID-19 case was reported in November 2020, lagging behind the rapid spread of the worldwide pandemic. For the survey, 30 percent of respondents were chosen from vulnerable households with a single parent, an income below the subsistence level, or a household with a disabled person. A total of 72 herder households and 40 vegetable grower households, 138 households living in *ger* district areas, 109 households living in apartments, and 4 agricultural farmers were surveyed. A majority (65.8 percent) of all respondent households were from rural areas, and the remaining minority (34.2 percent) were from the capital city. Vegetable households were randomly selected. The sampling of herder households was completed using the stratified random sampling method. The respective households in each strata, grouped by number of livestock (up to 1–100; 101–200; 201–300; 301–400; 401–500; 501–600; 601–700; 701–800; 801–900, and more than 900 livestock). Two to three herder households were selected from each livestock group from the sampled *soums*.

A semi-structured questionnaire with 205 questions was prepared, tested in advance, and improved accordingly. Each telephone survey took approximately one and a half hours and was conducted from June to July 2021 because the country was put under strict restrictions during that period. Eleven researchers of the Institute of Natural Resource and Agricultural Economics administered the semi-structured questionnaire survey. The author herself was one of the field researchers and enumerators of the study.

*4.3. Model Specification*

4.3.1. Principle Component Analysis (PCA)

In order to gain insight into the perception of the survey respondents with respect to their response, we performed a principal components analysis (PCA) on a number of variables from our survey that we felt could reflect this perception. The PCA was used to reduce the number of variables to a few factors that best explain the variation in the variables. These factors can be seen as the underlying perceptions leading to the answers to our survey questions. General techniques of PCA applied for analyses such as KMO, Bartlett's test of sphericity and anti-image options, and others [54].

The PCA procedure started from the correlation matrices between all the selected variables to check the strong or weak correlations. In addition, the determinant of the correlation matrix was checked against the value of 0.00001 to check multicollinearity/singularity. Data were extracted by the principal component analysis method and varimax, with the Kaiser normalization used as the rotation method.

4.3.2. Multiple Regression Analyses

In addition to the PCA, the following model is estimated for investigating the relationship between income change and other variables described in Section 5.3

$$Yi = \text{constant} + \beta_1 Pi + \beta_2 Hi + \beta_3 Vi + ei, \tag{1}$$

where:

$i$ = household type;
Y—income changes as a dependent variable.

The independent variables are:

P—set of variables concerning respondents' perception regarding different government policies;

H—set of variables to define respondents' experiences (age), education level, and family size;

V—set of variables indicating the risk and vulnerability as a direct impact of COVID-19.

Data cleaning and aggregation was completed first in MS Excel, and was then converted into SPSS and Stata file types for the regression analyses. In the regression analyses, the household type was the selection variable. Four farmer households were not representative of the total population in this area. Therefore, in the model, "farmer households" were excluded regarding their sample size.

## 5. Results and Discussion

### 5.1. Policy Responses Taken by the Government of Mongolia to Mitigate the Impact of COVID-19

The government of Mongolia (GoM) activated the State Emergency Committee in January 2020. As a result, various public health measures have delayed the first confirmed case of COVID-19 until 10 March 2020. The GoM immediately ensured that it was not further distributed, with no intensive care admissions or deaths until 10 November 2020, when the first internal case occurred.

Prevention messages focused on restricting international travel, suspending all training and educational activities, closing several types of businesses, and banning major public gatherings such as celebrations of national holidays [9,19].

Responding to the adverse socio-economic impacts of COVID-19, the GoM declared the following social protection measures: on 26 January 2020, 5.5 billion Mongolian tugriks (MNT) was allocated to COVID-19 prevention efforts [55]; under the government resolution #140, exemptions from social insurance contributions corporate and income taxes were waived from 1 April 2020 to 1 July 2021, applicable to all contributors (employers and employees) enrolled through their mandatory and voluntary schemes except for public sector workers and employees [56], and value-added tax exemptions for sugar, all kinds of rice, wheat seed, and wheat itself were established for the given period [57]; on 29 April 2020, the parliament of Mongolia approved a law to prevent and fight the COVID-19 pandemic and reduce the negative impact on the economy and society [58]; on 6 May 2020, provisions were made to increase child money payments and allowances for vulnerable people, the monthly benefit size (MNT 20,000 or USD 7.2 (at the official rate of the Mongol bank for the day)) was increased to MNT 100,000 (USD 36)—the approved Child Money Programme covers all children under 18 years old and is effective from April 2020 to 1 July 2022; a food stamp program for the poor was also approved: the monthly benefit size (MNT 16,000 for an adult and MNT 8000 for a child) was increased up to 32,000 (USD 11.5) and 16,000 (USD 5.8), respectively (1 May 2020 to 1 July 2022); and for social welfare pension, the monthly benefit size (MNT 188,000 or USD 67.6) was increased by MNT 100,000 (USD 36) [59].

Based on a FAO rapid assessment in April 2020, 62% of the herder households surveyed reported a 45% decrease in household income from the previous year, due to COVID-19 [60], and in a study by the NSO and World Bank in July 2020, 70% of herders' incomes declined steadily [17]. The significant reason for income loss was that cashmere prices fell, on average, by 45% in 2020, due to the Chinese border closure [17] and decreased global demand. Cashmere is the major income source for Mongolian herder households; 71% of livestock production income came from cashmere alone in 2018 [8]. Hence, the GoM subsidised a MNT 20,000 (USD 7.2) bonus per kilogram of cashmere for households with goats to substitute income loss from cashmere price falls (effective only for 2020) [59]. This was a sector-specific support.

On 18 November 2020, the Bank of Mongolia deferred mortgage repayments until 1 July 2022 [61]; on 2 December 2020, they discounted the price of improved fuel by 50–75 percent in the *ger* district of the capital city [62]. On 13 December 2020, the government decided to pay the electricity, heating, water, and garbage bills of all households (house/apartment up to 100 m$^2$) from 1 December 2020 [63]. The government extended this decision until 1 January 2022 [64]. The GoM continues to take the same measures as before, and more

policy measures have been approved. On 14 February 2021, a MNT 10 trillion economic stimulus package was approved [65], and, on 8 April 2021, the GoM released government resolution № 93, which approved a 300,000 MNT (or USD 107.9) cash transfer to every Mongolian citizen [66].

From the above measures, the specific decision direction for a rural household is the cashmere bonus program [61] and a soft loan with a 3% annual interest rate, with a total budget of MNT 500 billion to support agricultural production and increase herders' incomes and livelihoods within the framework of the MNT 10 trillion economic stimulus package [67]. Although the GoM announced various measures to control the outbreak, the situation worsened. The number of COVID-19 cases continued to rise, from a low of 349 on 18 May to 2746 on 18 June. Interviewees have expressed anxiety, worry, and restlessness due to worries of the highest infection numbers so far. In Mongolia, as of 28 September 2021, cases in the last seven days/1 million people was 5614; and deaths per 1 million people was 32 [68].

*5.2. Descriptive Results*

The survey focused on COVID-19's impact on livelihood, and the reaction to and effectiveness of the government response measures against the pandemic. The questionnaire includes basic family information (age, family size, education level, household members' vulnerability), income sources and production expenditure (all types of income sources and amounts, loans, income changes before and during COVID-19), livelihood capital (animal number, cultivated land area), significance level of the government's responses to COVID-19, food availability and price changes, shocks, risks, adaptability, and main challenges for livelihood and business before and during COVID-19.

The results of the descriptive statistics are presented in Table 1. The questionnaire included 362 households, of whom 46% were male respondents. The age of the participants ranged from 18 to 87 years, with an average of 45 years. The majority of respondents have completed high school education. The average number of household members was four. In terms of households' vulnerability, disabled household heads made up for 5% of the respondents, single-occupant households made up for 4.1%, women-headed households accounted for 9.4%, male-headed made up for 1.1%, households with a disabled person made up 1.7%, and households with incomes below the subsistence level accounted for 14%. Other households (64.7%) were classified in none of the above categories. Households with disabilities, female-headed households, and sick or disabled caregivers were often categorised as having incomes below the subsistence level. A high standard deviation of income level in both years indicates high income inequality within and between households groups.

**Table 1.** Descriptive statistics of the selected variables by household (HH) type.

| Variables | Apartment HH | | *Ger* District HH | | Herder HH | | Vegetable HH | | Farmer HH | |
|---|---|---|---|---|---|---|---|---|---|---|
| | **Mean** | **SD** | **Mean** | **SD** | **Mean** | **SD** | **Mean** | **SD** | **Mean** | **SD** |
| Age | 43 | 15 | 43 | 14 | 48 | 12 | 45 | 11 | 46 | 13 |
| Gender | 0.55 | 0.50 | 0.48 | 0.50 | 0.69 | 0.46 | 0.53 | 0.51 | 0.75 | 0.50 |
| Education level | 4.36 | 1.06 | 3.92 | 1.23 | 3.00 | 1.40 | 3.55 | 1.36 | 3.75 | 0.50 |
| Family Size | 3.60 | 1.195 | 4.00 | 1.626 | 4.42 | 1.659 | 3.95 | 1.679 | 5.00 | 0.816 |
| Any disabled or vulnerable family members in HH | 0.24 | 0.428 | 0.54 | 0.501 | 0.26 | 0.444 | 0.23 | 0.423 | 0.25 | 0.500 |
| Total Income 2019 (Mln.MNT) | 17.3 | 14.7 | 12.5 | 10.7 | 15.7 | 17.8 | 41.0 | 15.3 | 19.9 | 6.2 |
| Total Income 2020 (Mln.MNT) | 15.4 | 12.5 | 11.3 | 9.4 | 17.4 | 13.8 | 42.1 | 17.2 | 17.5 | 5.6 |

**Table 1.** *Cont.*

| Variables | Apartment HH | | Ger District HH | | Herder HH | | Vegetable HH | | Farmer HH | |
|---|---|---|---|---|---|---|---|---|---|---|
| | Mean | SD | Mean | SD | Mean | SD | Mean | SD | Mean | SD |
| Total Income_Change (Mln.MNT) | −1.9 | 8.5 | −1.2 | 5.6 | 1.6 | 14.5 | 1.1 | 26.9 | −2.4 | 2.7 |
| Government measures [1–5] | 3.4 | 1.4 | 3.6 | 1.3 | 3.3 | 1.4 | 3.0 | 1.4 | 3.4 | 0.7 |
| Interruption of main foodstuffs during COVID-19 [1–5] | 3.5 | 0.8 | 3.3 | 0.9 | 3.7 | 0.7 | 3.6 | 0.7 | 3.7 | 0.3 |
| Price increase during COVID-19 [1–5] | 3.6 | 1.0 | 3.7 | 1.0 | 3.5 | 1.2 | 3.3 | 1.1 | 3.6 | 0.3 |
| Valid N (listwise) | 109 | | 138 | | 72 | | 40 | | 4 | |

During lockdown, food and health services and some public services were allowed to continue their activities. Livestock and vegetable growing businesses were also not interrupted. Entertainment, restaurant, and hotel businesses shuttered and, even after lockdown, still followed some restrictions. Therefore, household income fell by 26.8 percent in 2020 compared to 2019, a year without COVID-19. The household types with declining incomes were farm households, apartment households, and *ger* district households. Most of these households were selected from the capital city; thus, their businesses were shut down completely several times, and their incomes declined. The income of vegetable grower households and herder households increased. This can be explained by the lesser impact of strict lockdowns on their business and benefits from the government measures.

Artmann, Tankam, and Zhang [69–71] found that if farmers are close to the market, they have a better opportunity to sell their products and lower their risks. Contrarily, in our case, farmers living close to the capital city lost their income entirely due to the lockdown. This could be explained by the similar research of Monirul et al. [72], who showed that small-scale farmers completely lost their income due to stopped domestic and international transportation and lockdowns. Herders, vegetable growers, rural households, and their businesses were not disrupted, and additionally, the government measures for every citizen increased their income. Food availability and its supply interruption happened in both rural and urban areas, as in other countries. The apartment and *ger* district households mainly answered that they had food shortages or that food was less available. This finding is in accordance with Kumar et al. [73], who found that for urban poor in India, international and regional market closures and an insufficient supply of goods resulted in a shortage of food. In addition, Ebata et al. [74] concluded that poor households of lower- and middle-income countries faced food and nutrition insecurity due to the COVID-19 crisis. Contrary, Raju et al. and Jia et al. [3,16] found that rural households are less impacted than urban households because of self-subsistence, as they are able to procure meat, milk, and vegetables on their own. Shammi et al. [29] found that product price hikes decreased daily consumption. The same trends happened in Mongolia; the increased price of eggs, meat, flour, and imported vegetables negatively influenced the apartment and *ger* district households. Traditionally, rural households' food basket are comprised of meat, milk, and flour products. According to our research, herders use milk and meat products from their farms for free, but the increased flour prices have strongly affected them.

We asked, "Which of the following measures taken by the government during COVID-19 was more important?" Participants rated the nine main measures taken by the government to stabilise household income and support livelihoods on a scale of 1 to 5. They highlighted three measures as the most impactful in their lives, which are as follows. (1) The electricity, water, and heating costs paid for by government option was chosen by 255 participants, and had the most votes (70.2%). The free water and central heating bills were only for apartment households. The electricity bill exemptions were for all households.

Herder households with no access to the central electricity grid have not benefitted from this measure. Price deductions of improved fuel by 50–75% (spring and winter seasons) were for the capital city's *ger* district households. This measure was most helpful during the complete lockdown, where businesses shuttered and household incomes were interrupted. (2) The 300,000 MNT (or USD 107.9) cash transfer for every citizen option was chosen by 255 people, or 70.2% of the participants. (3) The increasing the monthly child allowance from MNT 20,000 to MNT 100,000 (USD 7.2 to 36) option was chosen by 250 households (68.9%), which was a method that increased the monthly income of households immensely. The other measurements taken by the GoM were not intended for every citizen. Therefore, the number of responses that valued these measures as significant was fewer than the others (Table 2).

**Table 2.** Responses on how the measures taken by the government during COVID-19 are significant.

| N | Government Measures against Negative Impacts of COVID-19 | 1 = Not Significant at All (%) | 2 = Slightly Significant (%) | 3 = Moderately Significant (%) | 4 = Significant (%) | 5 = Very Significant (%) |
|---|---|---|---|---|---|---|
| 1 | Water, electricity, and heating cost paid by government | 6.3 | 1.1 | 8.5 | 13.2 | 70.2 |
| 2 | Cash transfer for every citizen MNT 300,000 (USD 107.9) | 2.5 | 2.2 | 9.6 | 15.4 | 70.2 |
| 3 | Child money amount increased (20,000 to MNT 100,000 or from USD 7.2 to 36) | 10.7 | 1.4 | 7.7 | 11.3 | 68.9 |
| 4 | MNT 1,000,000 (USD 360) per pensioned elder | 17.9 | 7.2 | 12.4 | 17.6 | 44.9 |
| 5 | Income tax waive and social insurance payment exemption | 23.4 | 6.1 | 18.7 | 27.0 | 24.8 |
| 6 | Deferred loan repayment | 27.8 | 12.9 | 19.3 | 19.0 | 20.9 |
| 7 | Soft loans for entities | 37.5 | 12.9 | 20.7 | 12.4 | 16.5 |
| 8 | Increased other state-funded benefits | 32.0 | 12.7 | 23.1 | 16.8 | 15.4 |
| 9 | Food vouchers for poor | 40.2 | 11.6 | 18.5 | 17.1 | 12.4 |

The following open question was asked to determine whether the change in income affected their purchases and the reasons: "If your purchases changed in 2020 compared to 2019, please state the reason for the change?". For this question, 26% answered that there were no changes, 16% answered that their purchases had declined, and 34% answered that prices had increased. Generally, the answers stated that the product variety decreased due to the lockdown, and that they had to purchase products at the *soum, aimag* centre at much higher prices.

To determine if, before COVID-19, the income and purchases of food and products were regular, we asked: "What are the main problems in your household's livelihood?" After COVID-19 started, the following problems suddenly increased: food shortage, unemployment, decreased income, debt, deteriorating children's education, and deteriorating health. In comparison, alcohol addicts decreased from 3.9% to 2.8%, and the tuition fee payment problem did not change, staying at 5–7.7%. Extensive bans on alcohol sales and services to prevent quarantine, domestic violence, and celebrations helped to reduce alcohol addiction and promote family peace. Respondents also said that because students study at home, they saved much money on pocket money, city transportation, food, transportation

between the countryside and city, and clothing. They only needed to pay tuition fees (Table 3).

**Table 3.** The main problems of the household before and during COVID-19 (%).

|  | Food Shortage | Unemployment | Decreased Income | Debt | Tuition Fee Payment | Deteriorating Children's Education | Deteriorating Health | Alcohol Addiction |
|---|---|---|---|---|---|---|---|---|
| During COVID-19 | 41.3 | 31.1 | 47.7 | 31.7 | 7.7 | 59.2 | 36.4 | 2.8 |
| Before COVID-19 | 0.6 | 1.7 | 2.5 | 4.1 | 5.0 | 0.6 | 1.1 | 3.9 |

N = 362.

Has anything positive happened to your family because of COVID-19? For this open question, 22% (80) of the participants answered that they spent more time with their families and became more familiar with the online environment, strengthening their immune system. In comparison, 27.8% (101) answered that there were no positive aspects. When asked about the negative impacts brought by COVID-19, 72 people said their income had decreased. Furthermore, other answers were that temporary jobs or businesses stopped, children's education deteriorated, they constantly experienced depression and worried all the time, the prices of medicine, food, and regular products increased, and family food consumption increased. The highest share is deteriorating children's education. Internet access and unfamiliarity with online education affected students' performances [21,22].

Moreover, people had to push back the importance of their other sicknesses, and it was harder to receive hospital services. Illness and death of a household member reduces household labour allocation and increases health or funeral expenditures [50]. For instance, when two people went to Ulaanbaatar city for a health inspection, they had to take the PCR test six times, creating more cost and delay. As another example, in a herder household, the husband was infected with COVID-19 and was sent to the hospital. His wife herded more than 400 animals alone. She expressed much worry about her husband's health, but could not visit him due to the duty of herding and the quarantine. Seven respondents expressed that they had to take food aid from their parents and relatives due to temporary job losses and a lack of income due to lockdown restrictions. The result is very similar to Waibel, H et al. [12], who found that, prior to COVID-19, urban households transferred some money to relatives or parents in the rural area; now, urban households are asking for food and other supports from rural ones.

The following question was asked: "what are the biggest challenges your household faced before and during COVID-19?" For this question, we listed the 14 most common challenges households face and presented them with the choice of 1 = Never, 2 = Seldom, 3 = Sometimes, 4 = Often, and 5 = Always. Before COVID-19, the top 3 answers under the choice 5 = Always were gasoline price (26.0%), shortage of labour (17.8%), and shortage of cash (17.3%); as for 4 = Often, gasoline price was once again ranked first, followed by shortage of cash and shortage of tools and equipment, for which the prices were listed. Among the answers for 3 = Sometimes, lack of information (32.0%), lack of spare parts, price (30.6%), and shortage of cash (27.6%) were ranked as the three most expressed answers. As for the challenges that were never present or almost non-existent, the following answers were chosen: high product loss (28.6%), fewer customers (26.7%), vehicle shortages and renting prices (21%), quarantine (64.9%), shortage of packaging (56.7%), and bureaucracy (42.9%).

Since the start of COVID-19, the impact of the previously listed challenges on household livelihood has changed. Challenges include quarantine (67.3%), followed by shortage of labour (35.1%) and vehicle shortages and renting prices (30.4%). For choice 4 = often, fewer customers (35.4%), gasoline price (35.0%), and quarantine (23.1%) were the most common answers.

The responses show that quarantine was the most pressing issue, affecting livelihoods the most. This result is similar to another study [35]. During the interviews, the rural participants mentioned that they used to go to the nearby *aimag* centre or city to buy cheap products, while at the same time selling their meat and dairy products and delivering winter food and clothes to their children. However, because of the quarantine, this became impossible to do. There was no choice left but to buy the scarcely available products in the *soum* at a higher price, and only sell their own products when a few traders and buyers come. Urban households reported that the supply of basic foodstuffs, such as eggs and meat, was interrupted, increasing price and scarcity. As a result, quarantine has had the most significant impact on the livelihoods of all the households. A similar result of the study by Alsuwailem et al. [17] proves that the food supply chain was interrupted around the globe. Like Mongolia, flour was the most demanded product in the UK [30] and Poland [75] during lockdown periods.

Households with livestock used their livestock for domestic consumption (preparing winter food for themselves and their children and relatives) in the fall and winter, earned some income by selling some, and maintained proper livestock in the herd. Overall, there has been very little change in the number of animals sold by herders over the past two years. Sales of horses and cattle declined, with sheep and goats selling slightly more than the previous year, with an average of 3 sheep and 18 goats. This trend may be due to the sharp decline in horse meat exports due to the closure of the Chinese border and the sharp decline in demand for beef from restaurants, catering businesses, schools, and kindergartens, which are the primary beef consumers. This is in line with the other studies by Lokonon, Dudek, and Siewak [49,75], who conclude that food market demand has changed globally, and that people prefer more readily available, easy-to-store and easy-to-transport food. Small animals are often used for food and household consumption, sometimes called "penny".

To find out whether the reasons for selling livestock have changed in 2019 and 2020, we asked: "what were the reasons for selling livestock and meat?" The options were "*dzud*", "dropped cashmere price", "tuition fee payment", "repair fences and houses", "to purchase car and motorcycle", "health and medical services", "for own consumption", "wedding and ceremony", "travel and leisure activity", "food", and "other purposes". Responses show that, since COVID-19, households have seen a sharp decline in the sale of livestock for travel, weddings, house repairs, and student-related expenses. For instance, the answer for "travel and leisure activity" decreased from 4 to 0, and for "wedding and ceremony" from 9 to 3. For "repair fences and houses", the answer decreased from 11 to 2, and for "tuition fee payment", it decreased from 5 to 1. Ellis and Møller et al. [51,52] found similar results in households coping with the unexpected shock. The priority strategy is to reduce consumption, and the final choice is to sell land and fixed assets.

On the other hand, the following answers increased: livestock selling with the purpose of "health and medical services" increased from 3 to 7, "for own consumption" increased from 4 to 14, and "food" increased from 2 to 7. The answers "*dzud*" had no noticeable change. The coincided shock of *dzud* and COVID-19 occurred in some rural areas, but both had a mild effect during the survey. Therefore, according to the research of Waibel, H et al., and Ellis, F [14,52], households were note forced to adopt intense actions against the incremental effects. Some studies by Luo et al. [2] have shown that COVID-19 had a significant impact on poor rural households. However, in those countries, rural household income is diverse, and off-farm employment contribution to the household income is high. Thus, when lockdown occurred, their income was disrupted. Most of the Mongolian herder households' income comes only from livestock [76].

### 5.3. Model Results

Using the techniques explained in Section 4.3, the following set of variables are defined:

Component 1 describes the government measurement to mitigate the negative impacts of COVID-19. Therefore, the government policy-related variables are the vital components of the data set. This component describes a perception of the government policy. If

government measurements are considered good, this could indicate that the household's livelihood will be secured, like examples from other countries [45]. Government measures of increased food vouchers for poor, increased social welfare pensions, deferred loan repayments, soft loans, and tax and social insurance discounts were all in component one for all four types of households. None of the other variables were included in this component for each household. Therefore, component 1 solely describes government measures.

Component 2 describes mainly the respondents' perception of the challenges faced due to COVID-19. To assess the impacts of COVID-19, the respondents were asked sets of questions. The aggregation of these questions is represented by the "COVID-19 impact factor", which includes numbers 0–11, indicating the number of measurements that must be taken due to COVID-19.

Component 3 describes mainly the risk and vulnerability to COVID-19. This classification has rapid vulnerability assessment questions and food shortage and price increase variables.

Concerning components 2 and 3, the variables are not as clearly identified as in component 1. Therefore, the follow-up regression analyses were run based on component 1, adding identified variables from components 2 and 3.

The principal component analysis (PCA) results identified two factors for the "*ger* district households" case, and three for the other types of household cases. Only the significant results of the PCA are presented in Table 4.

**Table 4.** Principal component analysis of survey respondents' perception by household types.

| Type of HHs | Apartment HHs | | | *Ger* District HHs | | Herders HHs | | | Vegetable Growers HHs | | |
|---|---|---|---|---|---|---|---|---|---|---|---|
| Components | 1 | 2 | 3 | 1 | 2 | 1 | 2 | 3 | 1 | 2 | 3 |
| Vulnerability assessment by extended code, double-counted more than one person in a HH | | | 0.52 | | −0.63 | | 0.60 | | | −0.63 | |
| COVID-19 impact: number of measures 0–11 | | | 0.88 | | 0.67 | | 0.53 | | | 0.54 | |
| Increase in the amount of child allowance | | | | | | | | −0.52 | 0.79 | | |
| Increased food vouchers for the poor | 0.80 | | | 0.74 | | 0.72 | | | | | 0.60 |
| Increased social welfare pensions | 0.75 | | | 0.68 | | 0.73 | | | 0.68 | | |
| The electricity bill was recovered fully by the government | | 0.60 | | | | 0.75 | | | 0.72 | | |
| Deferred loan repayment | 0.55 | | | 0.77 | | 0.69 | | | 0.78 | | |
| Soft loans | 0.77 | | | 0.81 | | 0.71 | | | | 0.55 | |
| Tax and social insurance discounts | 0.64 | | | 0.67 | | 0.80 | | | 0.71 | | |
| Distributed MNT 300 thousand to each | | 0.58 | | | 0.58 | | −0.80 | | | | |
| MNT 1 million was given to the retired people | | | | 0.58 | | | 0.51 | | | 0.61 | |

**Table 4.** *Cont.*

| Type of HHs | Apartment HHs | | | *Ger* District HHs | | Herders HHs | | | Vegetable Growers HHs | | |
|---|---|---|---|---|---|---|---|---|---|---|---|
| Components | 1 | 2 | 3 | 1 | 2 | 1 | 2 | 3 | 1 | 2 | 3 |
| Food shortage | | | −0.52 | | 0.60 | | | 0.78 | | | −0.62 |
| Price increase | | | 0.64 | | 0.64 | | | 0.50 | | | −0.66 |
| Before COVID-19, number of problematic issues | | −0.55 | | | | | | | | 0.52 | |
| During COVID-19, number of problematic issues | | | 0.83 | | 0.56 | | 0.64 | | 0.58 | −0.57 | |
| Variance explained (%) | 21.9 | 20.8 | 18.5 | 26.6 | 22.7 | 27.7 | 22.8 | 18.3 | 25.2 | 23.9 | 21.6 |
| KMO measure of sampling adequacy | | 0.628 | | | 0.723 | | 0.673 | | | 0.550 | |
| Bartlett's test of sphericity (Sig.) | | 0.000 | | | 0.000 | | 0.000 | | | 0.000 | |
| Number of observations | | 109 | | | 138 | | 72 | | | 40 | |

### 5.4. Regression Results

A multiple regression model using Equation (1) was conducted based on the two years of households' data presented in Table 5. The dependent variable is income changes

**Table 5.** Multiple regression model results.

| Variables | Apartment HH | *Ger* District HH | Herder HH | Vegetable Growing HH |
|---|---|---|---|---|
| (Constant) | 28.9 (3.22 ***) | 20.8 (3.31 ***) | 24.2 (1.44) | −66.9 (−2.9 ***) |
| Family Size | 0.156 (1.6 *) | 0.324 (3.59 ***) | 0.073 (0.51) | 0.212 (1.2) |
| COVID-19 impact; Y = 1, otherwise 0 | −0.176 (−1.73 *) | −0.206 (−2.41 ***) | −0.172 (−1.44) | 0.027 (0.18) |
| Age | −0.253 (−2.45 **) | −0.001 (0.012) | −0.169 (−1.27) | 0.55 (2.58 ***) |
| Profession_V | 0.24 (2.51 **) | −0.042 (−0.49) | −0.043 (−0.36) | 0.192 (1.2) |
| Increase in the amount of child allowance | −0.137 (−1.19) | −0.111 (−1.17) | −0.062 (−0.42) | 0.109 (0.58) |
| Food vouchers for poor | −0.001 (−0.005) | −0.142 (−1.37) | −0.049 (−0.32) | 0.652 (3.2 ***) |
| Deferred loan repayment | −0.119 (−1.04) | 0.153 (1.09) | −0.255 (−1.75 *) | 0.154 (0.80) |
| Soft loans for entities | 0.265 (2.06 **) | −0.005 (−0.04) | 0.073 (0.44) | 0.093 (0.43) |
| Tax and social insurance waive/exemption | 0.084 (0.79) | 0.033 (0.28) | 0.347 (2.04 **) | −0.105 (−0.51) |
| Cash transfer of MNT 300 thousand to each citizen | −0.046 (−0.45) | −0.185 (−2.03 **) | −0.027 (−0.22) | 0.055 (0.36) |
| MNT 1 million to the retired people | −0.146 (−1.31) | 0.078 (0.82) | 0193 (1.45) | 0.362 (1.62 *) |
| R | 49.4 | 43.4 | 49.1 | 65.9 |
| Adj. R Sqr. | 15.8 | 11.8 | 10.2 | 21.3 |
| Durbin–Watson | 2.3 | 1.86 | 1.6 | 2.07 |

Notes: *, **, and *** indicate significance at the 10%, 5%, and 1% levels, respectively.

Family size had a positive impact on income for the case of apartment households and *ger* district households. This can be explained by the fact that most government measures were based on individuals rather than households, e.g., child money, support for retired people, and food vouchers for the poor.

COVID-19 impact has had a negative impact on income changes, as expected, and this variable is significantly different from zero, again in the case of apartment and *ger* district households. This is a dummy variable, indicating whether or not households are affected by the COVID-19 in any form. This is consistent with the study of Du Zhi-Xiong et al. [77], who found that COVID-19 did not have direct effect on family farms, compared to other types of risks, and that household characteristics such as gender, age, and education were not variables related to the short-term impacts. Moreover, Dudek and Spiewak [75] found that COVID-19 affected the non-labour-intensive agriculture sector less. If the labour-intensive agriculture sector, such as fruit and vegetable production, strongly depends on foreign workers, the sector will be more affected. Nevertheless, it is an opportunity to change the food supply system to be more sustainable [75]. That might be the case for Mongolia, where the COVID-19 impact was less, or where its immediate impact timing is later.

Regression results show that there are almost no statistically significant factors influencing herder households. This is similar to the study of Du Zhi-Xiong et al. [77], who found that COVID-19 did not have a direct effect on family farms compared to other types of risks, and that household characteristics, such as gender, age, and education are not variables related to the short-term impacts.

A variable profession is significant for the apartment households; thus, its income from the impact of COVID-19 hits household heads with any profession less.

The government's supportive measures positively impact the income, i.e., food vouchers for vegetable grower households; soft loans for apartment households; tax and social insurance discounts for herder households; and MNT one million cash transfers to retired people for vegetable grower households. The supports, namely, food vouchers for the poor, deferred loan repayments, free electricity and utility bills, and soft loans for entities were implemented in many countries in the world [40–45]. In Malaysia, citizens assessed that an e-money transfer was the most effective measure taken by any government to initiate people-based economic growth [45].

Interestingly, there are some negative impacts from the government support on the income changes, i.e., the deferred loans for herder households and the distributed MNT 300 thousand to each person for *ger* district households. This result is not a meaningful explanation at the moment, which might need additional in-depth studies that focus on this subject specifically.

For developing countries, government support has helped maintain households' livelihoods significantly. For example, the Chinese government helped to sell fish and agricultural products; in India, a radio station was used as a means of connecting health services and the local government; in the Philippines, the government supported the flow of agriculture, food, and fishery products; Bangladesh announced incentive package for fisheries and agriculture sector [72]. The Mongolian government carried out similar activities. In the first half of 2020, the income of rural households fell sharply due to the market prices for cashmere [17,60]. Therefore, the Mongolian government responded by reimbursing MNT 20,000 (or USD 7) per kg of cashmere, which helped to prevent the deterioration of the livelihoods of herder households immensely. According to the study of FAO and UNDP [17,60], herder household's income would decline by 45% if the government was not supporting them through previously mentioned measurements.

## 6. Conclusions and Recommendations

Mongolia was one of the countries to take preventive measures against COVID 19, closing its borders, shutting down schools and businesses, restricting social gatherings, and banning international arrivals, allowing the country to self-isolate and avoid a more significant outbreak—measures that many countries have implemented. The research

aimed to assess the immediate effect of COVID-19 on the livelihood of different types of households and the impact of rapid government policy and measures to reduce the negative effect of COVID-19. In general, government policies and measures had reached their goal to protect households' livelihoods during COVID-19.

The study hypothesis is supported; thus, rural households, such as Mongolian herders and vegetable growers, are less affected by COVID-19 than households in the capital city and other urban areas. In other words, amongst the five types of households surveyed, rural households, such as herders and vegetable growers, were more resistant to the shock of COVID-19 than the other three types. The livestock and vegetable growing businesses are nearly not shuttered; providing the majority of their food on their own was essential for being more resistant. Apartment and *ger* district households benefitted the most from the government measures against COVID-19.

However, the government measures were successful in this time in Mongolia; it might be helpful to consider the measure's efficiency for a specific type of household. Therefore, for the hazard or shock preparedness, the government of Mongolia would have detailed policy instruments that better suit different types of households, instead of having generalized measures for all households. This message also applies to other developing countries, and will be particularly important if further restrictions are introduced on businesses in the future, with new negative consequences for households. Suppose new lockdowns and restrictions are not accompanied by adequate social programs: in that case, they are less likely to be widely accepted by society and will have significant consequences for households experiencing problems. For this reason, the results of this study may help to shape the right policy to mitigate the consequences of the COVID-19 pandemic.

It has to be acknowledged that there are several limitations to our paper. The study took place during the first wave of COVID-19, and the survey data is gathered before and during the COVID-19 situation. Therefore, the follow-up study should be useful for comparing the situation of post-COVID-19 or next waves. The government spent a large amount of money from the state budget and other funds; thus, neither the long-term burden on the country's economy or the indirect impacts on household livelihood are considered by this study. In this study, farmer households' participation was limited; therefore, the PCA and regression model results excluded farmer households.

**Author Contributions:** Conceptualization, G.G.; methodology, G.G. and K.P.; data curation, G.G.; writing—original draft preparation, G.G.; writing—review and editing, K.P.; funding acquisition, G.G. and K.P. All authors have read and agreed to the published version of the manuscript.

**Funding:** This research is supported by the Stipendium Hungaricum Scholarship Program, SHE-18862-002/2018.

**Informed Consent Statement:** Informed consent was obtained from all subjects involved in the study.

**Data Availability Statement:** Data is not publicly available, though the data may be made available on request from the corresponding author.

**Acknowledgments:** The authors would like to thank the Institute of Natural Resources and Agriculture Economics researcher in Mongolia for their tremendous support in gathering data from Mongolian households.

**Conflicts of Interest:** The authors declare no conflict of interest.

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
