# Peer review of "What Type of Households in Mongolia Are Most Hit by COVID-19?"

_sustainability, doi:10.3390/su14063557_

Round 1
Reviewer 1 Report
This is an extremely interesting article. I read it with the greatest pleasure. It contains many interesting threads related to poverty, living standards, or the socio-economic situation of the population during the COVID19 pandemic.
The article is very interestingly written, which in my opinion adds to its value.
Although a lot of literature has been analyzed I would like to ask you to take a look at some more valuable works of such authors as: Aleksandra Łuczak (Entropy and Sustainability), Michał Dudek and Ruta Śpiewak (Sustainability), Valensisi ( European Journal of Dev. Res.), Dickerson et al. (Wellcome Open Res.), Kalinowski (Social Policy Issues), Asmudson (J. of Anxiety Dis.), Cortes (Upjohn Institute Working Paper), Halamska (Rural and Villages). Some of them dealt with poverty, others with consumption or the situation of the population, including rural ones.
The only thing I would have to complain about is the ending. It is rather laconic. It should be slightly expanded. It could be supplemented with recommendations.
Please consider whether it is worth posing a research question in the introduction? Perhaps a hypothesis? Or perhaps a clear indication of the research problems?
Despite the comments and suggestions, I find this article extremely interesting.
Author Response
Dear Reviewer,
Thank you for your valuable review. We revised the manuscript accordingly. Please see the attachment and revised version of the manuscript.

Reviewer 2 Report
Summary
The study aims to explored and validate the measures taken by the Mongolian Government against the adverse effects of Covid-19, jeopardizing the livelihood of Mongolian Citizens. The study builds on a systematic literature review and a household survey. A total 362 households has been covered, taking into account different housing types namely apartment, district households, herder’s households, vegetable growing households and small farmer households. Principal component analysis and multiple regression analysis were used to facilitate the statistical analysis. Results focus on government policy; challenges due to Covid-19; and risk and vulnerability associated with Covid Results show that while herding households seem least affected compared to the other four types of households. The government measures were best suited to apartment and district households. Overall, rural households appear to be most resilient to Covid-19.
Dear authors
I read and appreciate your interesting manuscript. The content presented is timely and relevant, and presenting issues various countries are facing. Please find below my comment that are hopefully helpful to improve your manuscript.
P1-2: The introduction explains the Covid-19 sitution in Mogolia very clearly, however could you please include the various household types that have been mentioned and just briefly scratched when outlining the research gap. Please explain their relevance to the country. I think that would help readers not being familiar with Mongolia. Where are the specific differences between herders, vegetable growers and small farm household- in terms of potential risk and resiliency?
P3-4: On page 2 you indicate your first research question: How are the measures taken by the Government of Mongolia against Covid-19 affecting household livelihoods? In it’s current stage the conceptual framework and the literature are not answering well to that? Effects and consequences on livelihood are well presented, but any information on government responses is not present. Can you elaborate on this?
P4: Paragraph before the picture. This is extremely confusing. Are you describing what others have done? Or is this a description of your on literature review or a mix of both. In case you are describing there your own approach to the literature review following / adding on the work of someone else: this should be part of your method section.
Maybe at this point in the manuscript present the relevant results of the study- and shift technicalities of your proceeding in the method section. If you do so please include and justify inclusion and exclusion criteria of studies and how authors found agreement. If I misunderstood- please excuse and clarify.
P6: Can you please elaborate on how you recruited people? Were agriculture organization involved. Obviously via phone- Can you please elaborate and justify the appropriateness of your procedure of your criteria sampling?
P6: 206 questions- did you give people an incentive to participate? Or did you compensate their time? Please elaborate? Can you please elaborate how many peole participated in the survey in total- before you obtained 362 respondents? Drop out rate?
P15: Are there any best practice or policy recommendation that could be deducted from your work? Please address the government as well as other relevant stakeholders. Are there any suggestions for future research that can be deducted from your work? And lastly, please acknowledge limitation to your study.
Looking forward to see the revisited version of your work.
Author Response
Dear Reviewer,
Thank you for your valuable review. We revised the manuscript accordingly. Small and minuscule changes that came up during the proofreading were not tracked using track changes to make it easier to read. Please see the attachment and a revised version of the manuscript.

Reviewer 3 Report
Dear authors,
Congratulations for your work. The introduction and methods sections are well witten and the results are important for the understanding of Covid-19 Pandemic in the more remote areas such as rural Mongolia. However, a few changes are needed:
- The results should be better discussed, especially from the perspective of sustainability. Moreover, you should also try to highlight some of the lessons resulted from your research.
- The conclusions are rushed and do not properly present the results.
- You do not talk about the limitations of your study. For example, the data was collected using the telephone. But is it fully representative for the more remote areas and those who live bellow poverty line?
Author Response
Dear Reviewer,
Thank you for your valuable review. We revised the manuscript accordingly. Small and minuscule changes that came up during the English proofreading were not tracked using track changes to make it easier to read. Please see the attachment and a revised version of the manuscript.

Round 2
Reviewer 2 Report
All my comments have been thoroughly considered and implemented. I have no more reservation towards the paper. Was a pleasure to review your work
Reviewer 3 Report
Dear authors,
I am happy with the changes. I think that the quality of your manuscript has greatly improved. All the best!